# Prediction of the Current and Future Distributions of the Hessian Fly, *Mayetiola destructor* (Say), under Climatic Change in China

**DOI:** 10.3390/insects13111052

**Published:** 2022-11-15

**Authors:** Qi Ma, Jin-Long Guo, Yue Guo, Zhi Guo, Ping Lu, Xiang-Shun Hu, Hao Zhang, Tong-Xian Liu

**Affiliations:** 1State Key Laboratory of Crop Stress Biology for Arid Areas, College of Plant Protection, Northwest A&F University, Xianyang 712100, China; 2Yining Customs Technical Center, Yining 835008, China; 3Key Laboratory of Integrated Pest Management on Crops in Northwestern Loess Plateau, Ministry of Agriculture and Rural Affairs, China, Northwest A&F University, Xianyang 712100, China; 4Institute of Entomology, Guizhou University, Guiyang 550025, China

**Keywords:** Hessian fly, MaxEnt model, climate change, environmental variables, habitat distribution

## Abstract

**Simple Summary:**

The Hessian fly (*Mayetiola destructor* (Say)) is an important wheat pest worldwide and an invasive species in China. In this study, we used the MaxEnt model to predict the potentially suitable habitat of the Hessian fly in China under current and future climatic change conditions. The results showed that under the current climatic conditions, the potentially suitable habitats of the Hessian fly were mainly concentrated in central and eastern China, with an area of 96.27 × 10^4^ km^2^. With increasing global temperatures, most potential geographic distribution areas would expand in the future.

**Abstract:**

The Hessian fly, *Mayetiola destructor* (Say) (Diptera: Cecidomyiidae), is a destructive wheat pest worldwide and an important alien species in China. Based on 258 distribution records and nine environmental factors of the Hessian fly, we predicted the potential distribution area in China under three current and future (2050s and 2070s) climate change scenarios (RCP2.6, RCP4.5, and RCP8.5) via the optimized MaxEnt model. Under the current climate conditions, the suitable distribution areas of the Hessian fly in China were 25–48° N, 81–123° E, and the total highly suitable distribution area is approximately 9.63 × 10^5^ km^2^, accounting for 9.99% of the total national area. The highly suitable areas are mainly located in northern Xinjiang and central and eastern China. With the rising global temperatures, except for the high-suitable areas under the RCP8.5 scenario, most potential geographic distribution areas would expand in the future. The minimum temperature in February (tmin-2), precipitation in March (prec-3), maximum temperature in November (tmax-11), and precipitation seasonality (bio-15) are important factors that affect the potential geographic distribution of the Hessian fly. This study provides an important reference and empirical basis for management of the Hessian fly in the future.

## 1. Introduction

Climate change can directly affect the development, survival, and dispersal patterns of invasive species, possibly leading to an increased risk of biological invasions, pest outbreaks, range shifts, and habitat loss [1,2,3]. Climate warming is the main feature of climate change in the 21st century, and global temperatures are expected to rise between 1.4 and 5.8 °C by 2100, based on the 1990 baseline data [4]. Therefore, the impact of climate warming on invasive insects’ distribution has attracted international attention [5,6,7,8,9,10,11].

The Hessian fly, *Mayetiola destructor* (Say) (Diptera, Cecidomyiidae), is a destructive pest in many wheat production regions worldwide, causing up to 30–100% yield losses [12]. The most suitable host of *M. destructor* is the wheat plant (*Triticum* spp. L.). Besides wheat, *M. destructor* feeds on many cultivated and wild plant species belonging to more than 16 genera of Poaceae [13,14].

*M. destructor* originated in the Middle East and now occurs widely in North Africa, North America, Europe, Central Asia, New Zealand, and northwest China [12,15]. *M. destructor* may complete one to six generations per year, depending on the latitude and climate [16,17]. Adults usually lay eggs on their host leaves, and the neonate larvae move down tillers and establish feeding sites on the stem beneath the leaf sheath [18]. Although *M. destructor* has three instars [19], only the first and second instar larvae feed; they can affect the normal growth, jointing, and heading of wheat and even lead to death in severe cases [20]. The third instar and pupae develop in the second-instar exoskeleton, usually known as a puparium or “flaxseed” stage [17]. Adults do not feed and may live for 1–4 days [21]. *M. destructor* has a weak migratory ability and mainly relies on human activities for long-distance dispersal [18,22]. The developmental zero temperatures of the eggs, larvae, and pupae are 12.2, 1.6, and 1.6 °C, respectively; the optimum temperature for growth and adult emergence is 21.1 °C; when the temperature is higher than 26.7 °C, a large number of the third instar larvae enter dormancy [23]. Thus, it is a cool season pest. Additionally, *M. destructor* has the characteristics of facultative winter diapause and summer aestivation; the third instar larvae may aestivate over summer or diapause or non-diapause over winter, depending on their geographical locations [17,24]. Consequently, its life cycle ranges from 28 days to several years [25,26].

Due to its economic importance, *M. destructor* has been listed as an entry and domestic quarantine pest in China since 1954. However, in 1975, it was discovered in Huocheng County of the Ili prefecture (Ili), Xinjiang, China [27], and was then discovered in nine counties and cities in Ili and three counties in Bortala prefecture (Bortala) in 1983. Several outbreaks caused huge losses to local wheat yield [27,28]. By taking strict phytosanitary measures, *M. destructor* was strictly limited in its initial distribution regions in Xinjiang. Surprisingly, in recent years, *M. destructor* repeatedly erupted in Bortala [28]. Compared with the outbreaks in the early 1980s, the severe occurrence areas during the beginning of the 21st century expanded significantly to some higher altitude areas. Because the annual average temperature and precipitation in Xinjiang showed a fluctuating upward trend from 1961 to 2015 [29], we speculate that the expansion may be attributed to climate warming.

Species distribution models correlate the distribution data of a species with environmental variables (such as the climate, soil, vegetation, altitude, and host) that correspond to the distribution locations of the species, and the relationship between the geographical distribution of the species and environmental variables is analyzed to build models to predict the potential distribution areas of the species under different climatic conditions [30,31,32]. Currently, the main species distribution models include CLIMEX, GARP, DIVA-GIS, and MaxEnt. Different species distribution models perform differently in predicting the geographic distribution of a species [33]. The MaxEnt model is based on the maximum entropy theory. According to the known species distribution information and corresponding environmental variables, it simulates and analyzes the distribution state of the species when the entropy reaches the maximum under restricted conditions [34]. The MaxEnt model is less affected by sample bias, has strong stability and high accuracy, and is easy to operate [35]; thus, it has been used increasingly in recent years [36,37,38]. Some studies noted that the MaxEnt model is sensitive to sampling bias and is prone to overfitting under default parameters, which may result in unreliable prediction results [39]; therefore, it is necessary to optimize the model. The Akaike information criterion correction (AICc) index is usually used to measure the goodness of the statistical model fitting to avoid over-fitting of data [40].

Determining the geographic distribution of pests under future climate change conditions is key to developing long-term management strategies [41]. Wu et al. [42] used CLIMEX to predict the suitable habitat of *M. destructor* in China under the current climatic conditions, which provides a empirical basis for formulating prevention and control measures for *M. destructor*. However, it fails to reflect the changes in suitable habitats under future climatic conditions. Therefore, in this study we aimed to use ArcGIS and MaxEnt models in combination to (1) evaluate the main environmental factors that affect the geographical distribution of *M. destructor*, and (2) predict the range of the potentially suitable habitats of *M. destructor* in China under the current and future climate change scenarios, which will provide an empirical basis for the prevention and control of *M. destructor*.

## 2. Materials and Methods

### 2.1. Data and Processing of Species Presence Records

To obtain the occurrence record of *M. destructor*, we accessed two online databases, the Global Biodiversity Information Facility (https://www.gbif.org/, accessed on 15 January 2022) and the European and Mediterranean Plant Protection Organization (https://gd.eppo.int/, accessed on 15 January 2022). Other data were obtained by searching for published journal literature related to *M. destructor*. Additionally, a field investigation was conducted to obtain the actual distribution information in northern Xinjiang, China. The longitude and the latitude of each distribution point were determined through the Global Positioning System (https://www.gps-latitude-longitude.com/, accessed on 30 January 2022). Presence records with repeat latitude and longitude values, ambiguity, and missing distribution information were deleted according to the requirements of MaxEnt [43]. Finally, 259 distribution data points were selected to establish the prediction model (Appendix A).

The software, ArcGIS 10.4 (Esri, RedLands, CA, USA) was downloaded online (http://desktop.arcgis.com/zh-cn/, accessed on 10 November 2021), and the buffer analysis method in the software was adopted to screen and check the obtained species distribution points to avoid overfitting owing to large spatial correlation. The buffer diameter was set to 3 km, and when the distance between the distribution points was less than 6 km, only one of them was retained, resulting in 258 final distribution points (Figure 1). Then, 75% of the distribution points were randomly used as training data, and the residual 25% were set as the testing data to predict the potential distribution area of *M. destructor* in China.

### 2.2. Environmental Parameters

The method of obtaining and processing the environmental data used in this study was described by Li et al. [43]. Briefly, there were a total of 67 environmental factors, including 19 bioclimatic variables and 48 monthly climate factors (including the monthly maximum temperature, minimum temperature, average temperature, and precipitation), obtained from the Global Climate Data website (version 1.4, https://www.worldclim.org, accessed on 10 March 2022). The climate data were defined from 1970 to 1990. Four CO_2_ representative concentration pathways (RCPs) were published in the Fifth Assessment Report, which was issued by the Intergovernmental Panel on Climate Change [44]. Then, three gas emission scenarios, including RCP2.6, RCP4.5, and RCP8.5, which represent the minimum, medium, and maximum emission scenarios, respectively, were selected to simulate the habitat suitability distribution of *M. destructor* in the 2050s (2041–2060) and 2070s (2061–2080) [43]. In line with the IPCC Fifth Assessment Report, the general model CCSM4 was used to acquire the climate data. The spatial resolution was 2.5 arc-minutes (4.5 km^2^) for all the data. The specific environmental factors are shown in Table 1.

The software, MaxEnt (http://biodiversityinformatics.amnh.org/open_source/maxent, accessed on 10 November 2021, version 3.4.1) was used to eliminate the environmental factors whose contribution rate was less than 1.0%. To overcome the overfitting of the MaxEnt model we used the Pearson correlation coefficient analysis with the software, SPSS 19 (SPSS Inc., Chicago, IL, USA) by discarding highly auto-correlated environmental variables in the model [43]. When the correlation coefficient was greater than 0.9, there was a high correlation (Appendix A). Finally, nine key environmental factors were selected to model predictions of the possible areas of suitability. The Jackknife test was performed with the MaxEnt to evaluate the contribution of each environmental factor to the model construction [45], and the training scores “with only one variable”, “without a variable”, and “with all the variables” were calculated to judge the importance of the variables. The higher the score is, the greater its importance will be.

### 2.3. MaxEnt Model Construction and Parameter Optimization

To optimize the MaxEnt model, the ENMeval data R package was used to adjust the parameters of regulation multiplier (RM) and feature combination (FC), and the minimum value of AICc index was determined as the optimal setting [11,46], establishing the final model. The smaller the AICc value, the lower the complexity and the more reliable the model [47]. AICc = 0 means the best-performing models [40].

The *M*. *destructor* default fitness index of the prediction results from the MaxEnt model ranged from 0 to 1; the closer the values are to 1, the higher the possibility of the species’ presence. Jenks module in ArcGIS was used to evaluate the grades of suitable areas, and the classification criteria for the grades are as follows: unsuitable (*p* ≤ 0.16), low-suitable (0.16 < *p* ≤ 0.29), moderate-suitable (0.29 < *p* ≤ 0.81), and high-suitable areas (*p* > 0.81).

The receiver operating characteristic (ROC) curve and the area under the curve (AUC) were used to test the accuracy of the prediction potential for the distribution of the species. The AUC value has high reliability because it is not affected by the threshold [45]. Theoretically, the AUC value ranges from 0 to 1, and the closer the AUC value is to 1, the greater the correlation between the environmental variables and distribution model and the higher the accuracy of the prediction results [48]. The grade of simulation prediction specific standard was as follows: AUC ≤ 0.60, fail; 0.60 < AUC ≤ 0.70, poor; 0.70 < AUC ≤ 0.80, fair; 0.80 < AUC ≤ 0.90, good; 0.90 < AUC ≤ 1.0, excellent [49].

## 3. Results

### 3.1. Model Performance

Based on 258 current distribution data sets and nine environmental factors, the potential geographical distribution of *M*. *destructor* in China was simulated in MaxEnt with the default parameters (RM = 1, FC = LQHPT) and the optimized parameters (RM = 2, FC = QH), respectively. As a result, the AUC values from the default and optimized parameters were 0.923 and 0.933, respectively (Figure 2), reaching an excellent level. In addition, the delta AICc was 0 under optimized parameters but 43.941 under default parameters, indicating that the optimized model performance was significantly improved, reducing the complexity and avoiding overfitting. Therefore, the prediction result of the optimized MaxEnt model was accurate and available. 

### 3.2. Dominant Environmental Factors Affecting Distribution

According to the contribution rate, nine environmental factors greatly influenced the distribution range of *M*. *destructor* in the MaxEnt model (Figure 3). The nine factors had a cumulative contribution of 99.8%, and their contribution rates from high to low were as follows: tmin-2 (27.6%), prec-3 (22%), tmax-11 (20.5%), tmax-6 (10%), bio-15 (6.7%), bio-2 (5%), prec-11 (3.2%), bio-13 (2.5%), and bio-7 (2.3%).

The results of the Jackknife test showed that four key factors with the greatest influence were tmax-11, tmin-2, prec-3, and bio-15 (Figure 4).

The response curve of the four key factors is shown in Figure 5. According to the curve of different factors, when the probability of *M*. *destructor* distribution was ≥0.16, the suitability grade was a low-suitable distribution, tmax-11 was −2.01 to 24.92 °C, tmin-2 was −21.18–11.25 °C, prec-3 was 11.09–253.93 mm, bio-15 ranged from 0 to 93.34. When the probability of *M. destructor* distribution was ≥0.29, and the suitability grade was a moderate-suitable distribution, tmax-11 was 1.3–23.45 °C, and the optimum temperature was 20.05 °C; tmin-2 was −17.26–8.82 °C, and the optimum temperature was −2.39 °C; prec-3 was 23.13–203.67 mm, and the peak was reached when the rainfall reached approximately 72.69 mm. In addition, bio-15 ranged from 0 to 61.19. 

### 3.3. Predicting the Current Distribution of M. destructor in China

Under the current climatic conditions, the potential geographic distribution of *M*. *destructor* in China is shown in Figure 6. The suitable habitat area was 5.46 × 10^6^ km^2^, accounting for approximately 56.64% of the total area of the country, located at 25–48° N, 81–123° E, mainly in the central and eastern regions. Specifically, the high-suitable area was 9.63 × 10^5^ km^2^, accounting for 9.99% of the total area of China, and it was mainly distributed in northern Xinjiang, Hunan, Hubei, Chongqing, Guizhou, Sichuan, Jiangxi, Anhui, Jiangsu, Zhejiang, Shanghai, Henan, Fujian, Taiwan, and Shaanxi province. In contrast, the moderate-suitable area was 1.52 × 10^6^ km^2^, accounting for 15.76% of the total area of China, and the low-suitable area was 2.98 × 10^6^ km^2^, accounting for 30.89% of the total area of China.

Considering that Xinjiang is the only distribution area in China, the potential distribution range of *M*. *destructor* was especially analyzed, and the potential distribution of *M*. *destructor* in Xinjiang is shown in Figure 7. Under the current climatic conditions, the suitable distribution area was 77.12 km^2^, accounting for approximately 46.46% of Xinjiang’s total area. The high-suitable area accounted for approximately 19.62% of the total suitable areas in Xinjiang and is mainly located in Ili Prefecture, Bortala Prefecture, Tacheng Prefecture, Karamay City, Altay Prefecture, and Changji Prefecture.

### 3.4. Predicting the Future Suitable Climatic Distribution of M. destructor in China

Compared with the potentially suitable areas of *M. destructor* in China under current climate conditions, the future suitable area of the fly showed an increasing trend under the three scenarios, RCP2.6, RCP4.5, and RCP8.5 (Figure 8 and Figure 9). The main changes are as follows:

Under the RCP2.6 scenario, the total suitable area would decrease by 6.24% in the 2050s (2041–2060) while increasing by 13.49% in the 2070s (2061–2080). Similarly, the high-suitable area would decrease by 29.4% in the 2050s but expand by 9.08% in the 2070s, and the low-suitable area would decrease by 5.51% in the 2050s and expand by 5.95% in the 2070s. In contrast, the moderate-suitable area would increase by 7.03% and 31.07% in the 2050s and 2070s, respectively (Figure 9a).

Under the RCP4.5 scenario, the total suitable area would increase by 8.58% and 16.49% in the 2050s and 2070s, respectively. Similar to the total suitable area, the high-suitable area would increase by 5.29% and 36.16% in the 2050s and 2070s, respectively; and the moderate-suitable area would increase by 34.6% and 25.86% in the 2050s and 2070s, respectively. Conversely, the low-suitable area would decrease by 3.64% in the 2050s but increase by 5.29% in the 2070s (Figure 9b).

Under the RCP8.5 scenario, the total suitable areas would decrease by 4.04% in the 2050s while increasing by 1.65% in the 2070s. In the 2050s and 2070s, the high-suitable area decreases by 43.63% and 71.97%, respectively. Similarly, the moderate-suitable area increases by 24.87% and 11.1%, respectively. The low-suitable areas would decrease by 5.99% in the 2050s and expand by 20.64% in the 2070s (Figure 9c).

The potentially suitable area for *M. destructor* in Xinjiang in the future is shown in Figure 10. Compared with the current conditions, the total suitable area of *M. destructor* in Xinjiang tended to decrease in the 2050s but increase in the 2070s under the RCP 2.6 and RCP 8.5 scenarios, whereas the suitable area would increase in the 2050s but decrease in the 2070s under the RCP4.5 scenario (Figure 11). The main results were as follows:

Under the RCP2.6 scenario, the high-suitable area of *M. destructor* in Xinjiang increases by 46.78% and 30.69% in the 2050s and 2070s, respectively; the moderate-suitable areas increases by 3.74% and 12.75%, while the low suitable area decreases by 27.82% in the 2050s and expands by 19.59% in the 2070s (Figure 11a).

Under the RCP4.5 scenario, the high-suitable area of *M. destructor* in Xinjiang would decrease by 11.26% in the 2050s and increase by 17.36% in the 2070s. The moderate-suitable area would increase by 79.21% and 37.07% in the 2050s and 2070s, respectively. In contrast, the areas of low suitable areas would decrease by 14% and 18.02%, respectively (Figure 11b).

Under the RCP8.5 scenario, the high-suitable area of *M. destructor* in Xinjiang would increase by 18.3% and 18.67% in the 2050s and 2070s, respectively. The area of moderate-suitable areas would increase by 10.18% and 48.07%, respectively. However, the area of low suitable areas would decrease by 57.43% and 9.59% in the 2050s and 2070s, respectively (Figure 11c).

In the future, as the grade of suitable habitat increases according to the prediction results, the risk of some areas being invaded would increase. For example, part of Kizilsu Kyrgyz Prefecture (Wuqia and Artush counties), Aksu Prefecture (Baicheng and Wensu counties), and Kashi Prefecture (Shufu County) would change from the moderate-suitable to high-suitable grade. In contrast, some high-suitable areas, e.g., part of the area in Tacheng prefecture, would be downgraded to moderate-suitable owing to climatic warming, especially under the RCP4.5 and RCP8.5 scenarios. In addition, climatic warming would result in the southern boundary of distribution in Bayingolin County gradually moving northward under the RCP4.5 and RCP8.5 scenarios (Figure 10).

## 4. Discussion

### 4.1. The MaxEnt Model

Prevention of invasive alien pests is more economic than post-introduction pest management [50]. While the scope of invasion of invasive species in the future is difficult to predict [11], niche models are increasingly being used to predict the potential habitat of invasive species, such as CLIMEX, GARP, DIVA-GIS, and MaxEnt [33]. Compared with other models, the MaxEnt model has stronger performance and higher accuracy [51,52], and the best prediction results when the ROC curve is used to evaluate the reliability of the model [48,49]. In this study, we used an optimized MaxEnt model to predict the potential distribution range of *M. destructor* under current and future climate conditions. The AUC value of the optimized model was 0.933, reaching an excellent level. This suggests that the predicted and actual geographic distribution of *M. destructor* were highly similar, and the prediction results were highly reliable and accurate.

### 4.2. Environmental Factors Affecting the Potential Distribution of M. destructor

The distribution of a species is highly susceptible to environmental influences, and environmental conditions directly or indirectly affect the physiological and ecological functions, limiting the distribution of the species [53,54]. In this study, temperature, including the highest temperature in November (tamx-11) and the lowest temperature in February (tmin-2), and precipitation, including the precipitation in March (prec-3) and the precipitation variation coefficient (bio15), were the main environmental factors affecting the distribution of *M. destructor* (Figure 4). Previous studies revealed that ambient temperature and humidity were the two most important environmental factors affecting *M. destructor* [26,55,56,57], suggesting that the prediction result related to the environmental factors is credible in this study. Furthermore, the two temperature factors correspond to the overwintering period of *M. destructor*, while the precipitation in March corresponds to the pupae development stage of the overwintering generation in Xinjiang and most other potential distribution areas [17,24,27].

In this study, tmax-11 ranged from −2.01 to 24.92 °C, which is suitable for *M. destructor*. When the temperature was 20.05 °C, the probability of distribution of *M. destructor* reached a maximum value, indicating the optimum temperature for *M. destructor* development (Figure 5a). Tmin-2 was −21.18 to 11.25 °C, and the optimum temperature was −2.39 °C, favorable to the presence of *M. destructor*. Previous studies support our predictions. Foster and Taylor [23] suggested that the optimum temperature for *M. destructor* growth and development is 21.1 °C. McColloch [25] reported that the third instar larvae, the major stage to overwinter in most of the occurrence areas, can survive extreme ambient temperatures ranging from −26.7 to 37.8 °C. 

Ambient humidity can significantly affect the hatching of eggs, the survival, aestivation, and diapause of larvae, and the adult emergence of *M. destructor* [26,57]. Zhang [27] found that the rainfall in mid-March was crucial for the occurrence of the spring populations of *M. destructor* in Xinjiang, China. It was observed that the precipitation in March when it is >253.93 mm, or <11.09 mm are unsuitable for *M. destructor*; our results are consistent with the previous study.

### 4.3. The Potential Distribution of M. destructor in China and in Xinjiang Only

In this study, the range of suitable habitats of *M. destructor* in China were 25–48° N, 81–123° E, with an area of 5.46 × 10^6^ km^2^. This accounts for about 56.64% of China’s total area, of which the proportions of high-, moderate-, and low-suitable areas were 9.99%, 15.76%, and 30.89%, respectively. Compared to the results predicted by Wu et al. [42] using the CLIMEX model, the overall distribution of the suitable areas was consistent, and the high-suitable areas were concentrated in the central and eastern regions of China. However, there are some differences between the two prediction results, mainly in the following aspects: (1) the area of the suitable habitats in this study was slightly smaller than that of Wu et al. [42], and (2) the grades of some of the suitable areas were different. For example, Wu et al. [42] found that Tibet, most of Inner Mongolia, and the whole of Heilongjiang were low-suitable areas for *M. destructor*, and both the existing distribution and potentially suitable areas in Xinjiang were low-suitable areas. In contrast, this study showed that the northeastern part of Inner Mongolia, northern Heilongjiang, and most of Tibet were non-suitable areas, while Xinjiang, Bortala, Ili, Tacheng, and other areas were high-suitable areas for *M. destructor* (Figure 7). The difference in the prediction results of the two models may be related to the difference in parameter selection. Nevertheless, according to some studies, *M. destructor* has repeatedly erupted in the local areas of Bortala in Xinjiang, such as in Wenquan County and Bole City [28], indicating that these areas are likely high-suitable areas for *M. destructor*, which is consistent with our predictions.

In addition, under the three RCP scenarios, between 2041–2060 and 2061–2080, the suitable occurrence range of *M. destructor* increases or decreases to different degrees. This indicates that climate warming may change the suitable habitat of *M. destructor* in China. Under the RCP2.6 and RCP4.5 scenarios, the range of the high-suitable area of *M. destructor* showed a fluctuating upward trend, while under the RCP8.5 scenario, the area of the high-suitable area significantly decreased (Figure 9). In Xinjiang, under the three scenarios, the high- and moderate-suitable areas of *M. destructor* showed an increasing trend, and the areas of Tacheng, Bortala, and Ili adjacent to Kazakhstan were always high-suitable areas (Figure 10), possibly owing to the similar climatic conditions of these regions to Kazakhstan.

According to our predictions, under the current climatic conditions, except for Xinjiang, the potentially suitable areas of *M. destructor* were larger in Hunan, Chongqing, Hubei, Anhui, Shanghai, Zhejiang, Jiangxi, Guizhou, Henan, and Jiangsu, among which Henan, Anhui, and Jiangsu are the main wheat producing areas in China, accounting for 23.93%, 12.09%, and 10% of the total wheat planting area in China, respectively. Thus, once *M. destructor* is introduced into these areas, it will pose a huge threat to wheat production. Therefore, phytosanitary measures should be strengthened, such as prohibiting the export of wheat crop straws, bedding, or fillers from the epidemic areas to prevent the spread of *M. destructor* from Xinjiang.

According to the National Bureau of Statistics of China (http://www.stats.gov.cn/, accessed on 1 October 2022), the annual wheat planting area in China is approximately 2.38 × 10^5^ km^2^. Under the current climatic conditions, the potential distribution area of *M. destructor* in China is much larger than the annual wheat planting area. Moreover, many of these potentially suitable areas are natural pastoral areas where various Poaceae forages grow, such as *Agropyron repens*, *A. gaertn*, and *Echinochloa crusgalli*, all of which are wild hosts of *M. destructor* [27]. Prestidge [58] reported that *M. destructor* threatened the prairie grass *Bromus willdenowii*, an important forage grass in the North Island of New Zealand. Therefore, once *M. destructor* is introduced into these suitable areas, it may not only affect the growth of forage grass but could also affect the development of animal husbandry. It may also become a source of insects for the wheat planting areas adjacent to the pastoral areas.

Currently, no *M. destructor* have been found in central and eastern China, the major wheat concentration planting area in China [59]. The reason may be that the adult has a short lifespan, usually only 1–4 days [21,60], and its migration ability is poor [61]. Also, between the distribution area of Xinjiang and the non-occupied area, central and eastern regions of China, there is a vast unsuitable area, for example, southeastern Xinjiang, western Qinghai, and most of the Qinghai-Tibet Plateau, where there are many natural barriers formed by high mountains, making it difficult for *M. destructor* to spread to the central and eastern wheat areas by natural diffusion. Additionally, at present, the distribution area of *M. destructor* in Xinjiang is still limited to parts of Bortala and Ili Prefecture; the main reason may be the relatively effective control measures adopted locally.

Although MaxEnt has evident advantages for niche models, there were limitations to the model in this study. Sixty-seven environmental variables related to temperature and humidity were used in the study; however, the other environmental factors that affect the distribution of *M. destructor*, such as the host range, natural predators, topography, and altitude, were not considered. In addition, the feedback curve reflects the role of a single environmental variable, while the behavior of insects is affected by multiple environmental variables. Therefore, in future studies, the impact of multiple complex environmental factors on *M. destructor* should be comprehensively considered to improve the accuracy of the prediction results.

## 5. Conclusions

In this study, the MaxEnt model and ArcGIS software were used to simulate the potential distribution areas of *M. destructor* in China under current and future climate conditions predicted from three climate scenarios (RCP2.6, RCP4.5, and RCP8.5). Under current climate conditions, the range of potential distribution areas of *M. destructor* were 25–48° N and 81–123° E, mainly in northern Xinjiang, central and eastern China. In the future (2050s and 2070s), the potential distribution areas will expand under RCP2.6 and RCP4.5 scenarios but reduce for RCP8.5 scenarios. The important environmental factors that affect *M. destructor* include the maximum temperature in November (tmax-11), minimum temperature in February (tmin-2), precipitation in March (prec-3), and precipitation seasonality (bio-15).

## Figures and Tables

**Figure 1 insects-13-01052-f001:**
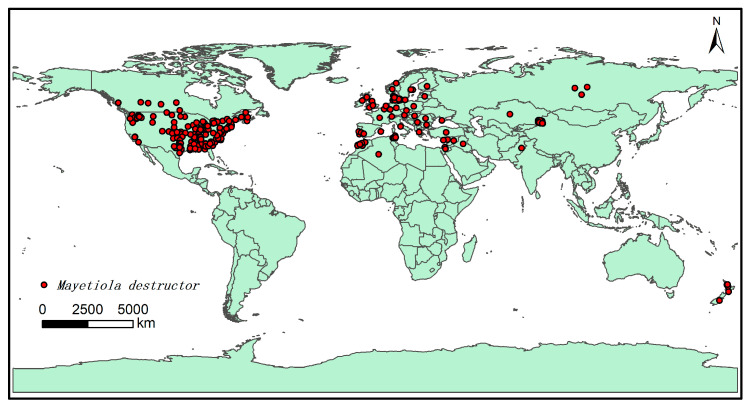
Worldwide sampling map of *Mayetiola destructor*. The map was obtained from the National Basic Geographic Information website (http://nfgis.nsdi.gov.cn, accessed on 30 January 2022). The sampling points are from databases, literature, and actual survey data, and they were drawn using ArcGIS.

**Figure 2 insects-13-01052-f002:**
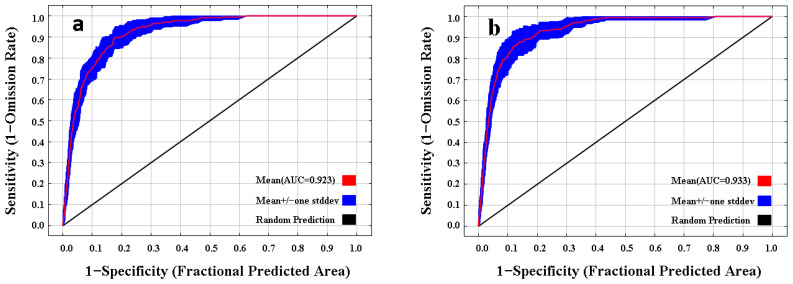
ROC curve and AUC values for the initial model (**a**) and the final model (**b**).

**Figure 3 insects-13-01052-f003:**
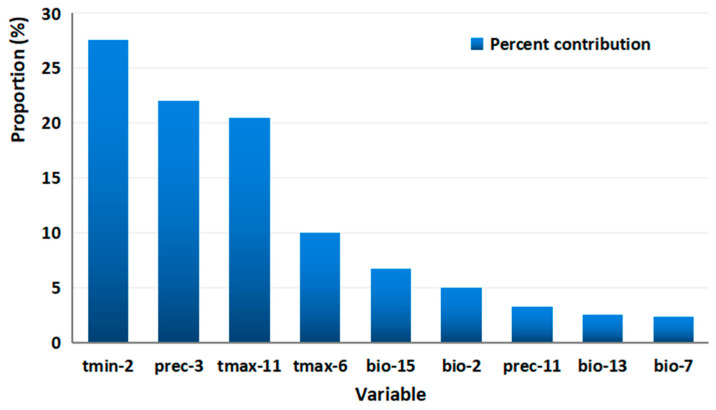
Percentage contribution of the nine variables. *x*-axis variables represent the minimum temperature in February (tmin-2), precipitation in March (prec-3), maximum temperature in November (tmax-11), maximum temperature in June (tmax-6), precipitation seasonality (bio-15), mean diurnal range (bio-2), precipitation in November (prec-11), precipitation of the wettest month (bio-13), temperature annual range (bio-7).

**Figure 4 insects-13-01052-f004:**
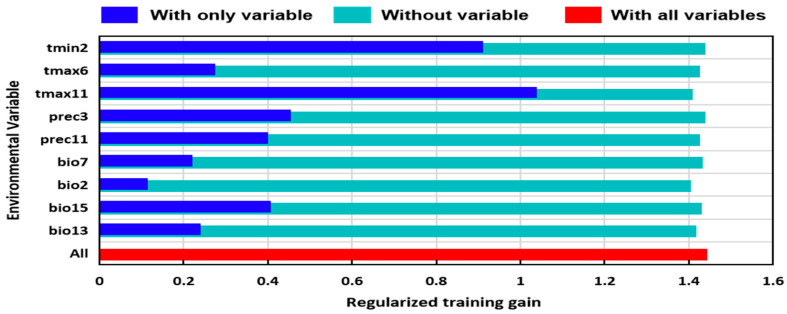
Relative importance of nine environmental variables for the distribution of *M*. *destructor* determined by the Jackknife test.

**Figure 5 insects-13-01052-f005:**
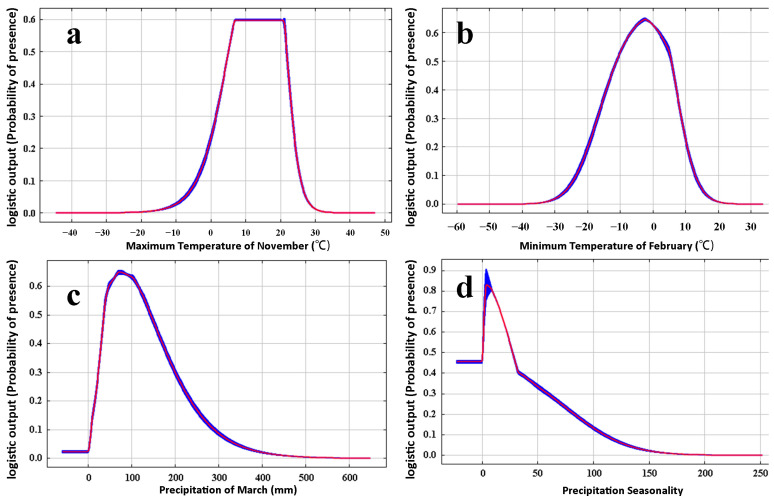
Response curves of *M*. *destructor* to various environmental factors in the MaxEnt model. (**a**) Maximum temperature of November (tmax-11), (**b**) minimum temperature of February (tmin-2), (**c**) precipitation of March (prec-3), (**d**) precipitation seasonality (bio-15). Red curves show the mean response and blue margins are ± SD calculated over 10 replicates.

**Figure 6 insects-13-01052-f006:**
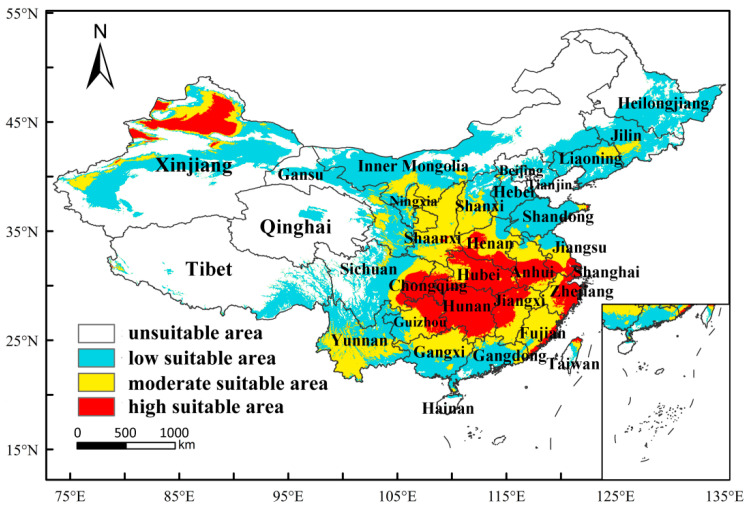
Predicted potential distribution of *M*. *destructor* in China under the current climatic conditions.

**Figure 7 insects-13-01052-f007:**
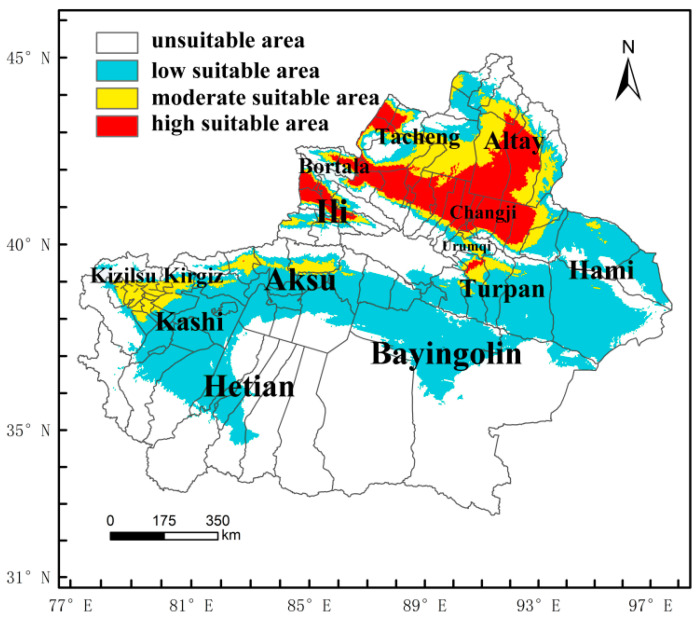
Predicted potential distribution of *M*. *destructor* in Xinjiang, China, under the current climatic conditions.

**Figure 8 insects-13-01052-f008:**
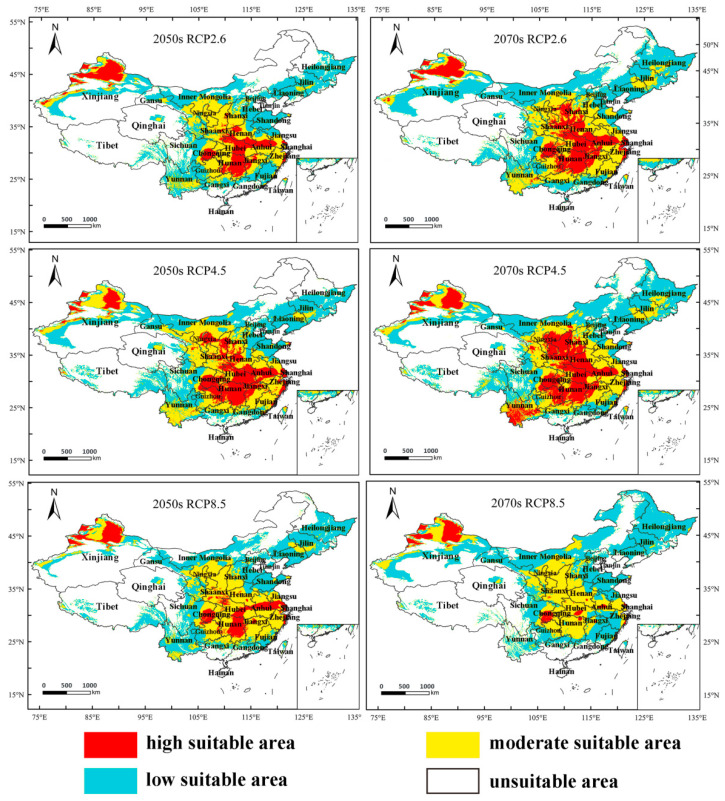
The predicted potential distribution area of *M*. *destructor* in China under three different climate scenarios in the future.

**Figure 9 insects-13-01052-f009:**
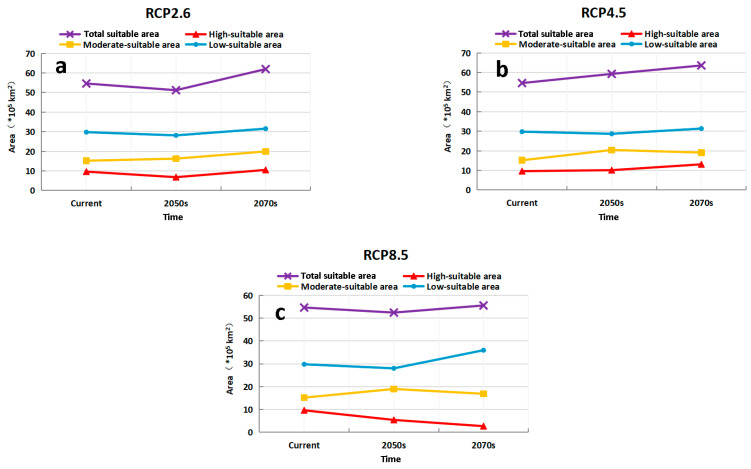
Predicted trend of the grades of habitat suitability for *M. destructor* in China under different future climate scenarios. (**a**) area in different periods under RCP2.6 (the minimum gas emission scenarios), (**b**) area in different periods under RCP4.5 (the medium gas emission scenarios), (**c**) area in different periods under RCP8.5 (the maximum gas emission scenarios).

**Figure 10 insects-13-01052-f010:**
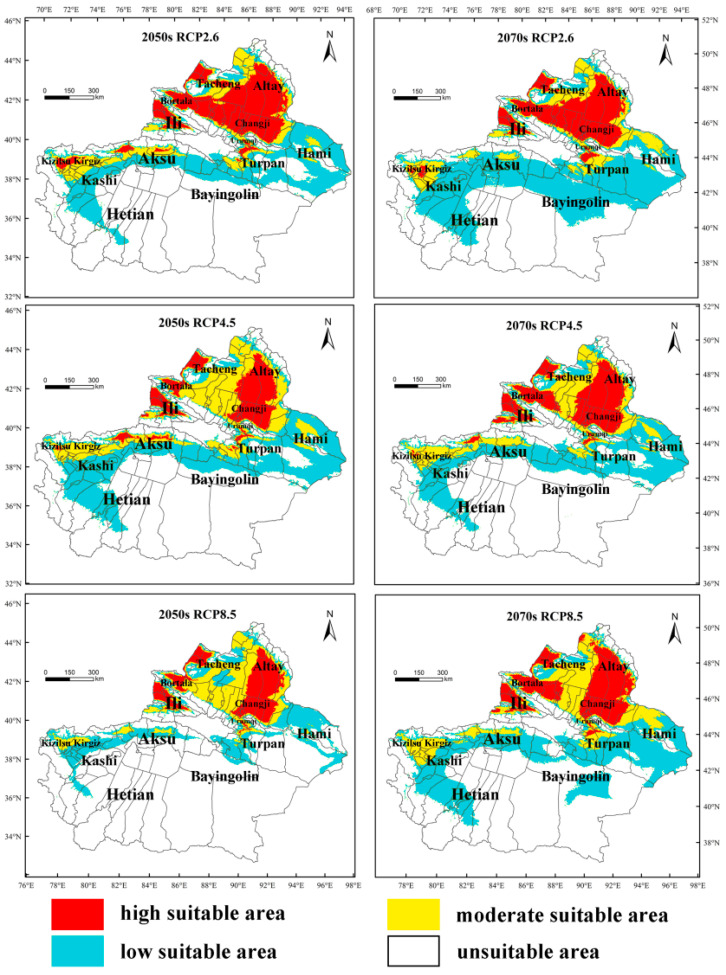
The predicted potential distribution area of *M. destructor* in Xinjiang, China, under three different climate scenarios in the future.

**Figure 11 insects-13-01052-f011:**
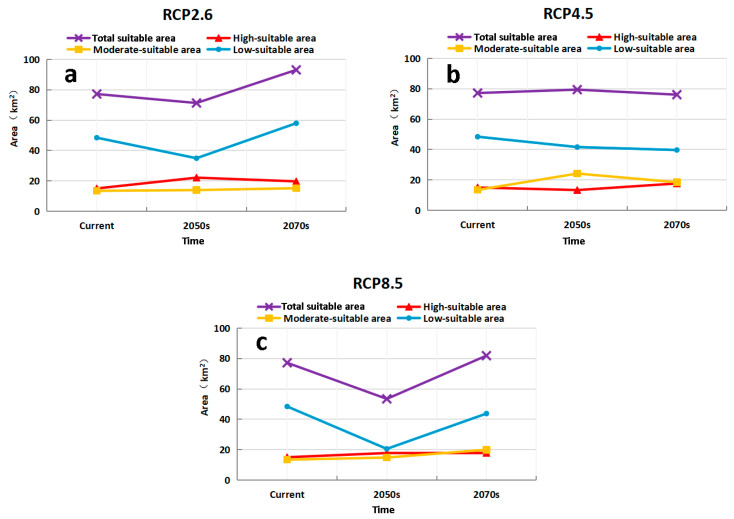
Variation trend of the grades of habitat suitability for *M. destructor* in Xinjiang under three climatic scenarios.(**a**) area in different periods under RCP2.6 (the minimum gas emission scenarios), (**b**) area in different periods under RCP4.5 (the medium gas emission scenarios), (**c**) area in different periods under RCP8.5 (the maximum gas emission scenarios).

**Table 1 insects-13-01052-t001:** Environmental factors for the potential geographic distribution of *Mayetiola destructor*.

Code	Environmental Factors
Bio1	Annual mean temperature (°C)
Bio2	Mean diurnal range (°C) (Monthly mean [max temp − min temp])
Bio3	Isothermality ([BIO2/BIO7] [×100])
Bio4	Temperature seasonality (standard deviation ×100) (°C)
Bio5	Maximum temperature of the warmest month (°C)
Bio6	Minimum temperature of the coldest month (°C)
Bio7	Temperature annual range (BIO5-BIO6) (°C)
Bio8	Mean temperature of wettest quarter (°C)
Bio9	Mean temperature of driest quarter (°C)
Bio10	Mean temperature of warmest quarter (°C)
Bio11	Mean temperature of coldest quarter (°C)
Bio12	Annual precipitation (mm)
Bio13	Precipitation of the wettest month (mm)
Bio14	Precipitation of the driest month (mm)
Bio15	Precipitation seasonality (coefficient of variation)
Bio16	Precipitation of the wettest quarter (mm)
Bio17	Precipitation of the driest quarter (mm)
Bio18	Precipitation of the warmest quarter (mm)
Bio19	Precipitation of the coldest quarter (mm)
Tmin	Average monthly minimum temperature (°C)
Tmax	Average monthly maximum temperature (°C)
Tmean	Average monthly mean temperature (°C)
Prec	Average monthly precipitation (mm)

## Data Availability

The data are included in the article. For the data provided in this study, see the Section 2.1 and Section 2.2 in the text.

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
