# Peer review of "Prediction of the Current and Future Distributions of the Hessian Fly, Mayetiola destructor (Say), under Climatic Change in China"

_insects, 2022, doi:10.3390/insects13111052_

Round 1

Reviewer 1 Report

Title: Prediction of the Current and Future Distributions of the Hessian Fly, Mayetiola destructor (Say), under Climatic Change in China

This paper is aimed to predict current and future distribution of the destructive wheat pest Hessian fly. I think that this paper is properly written with good results using Mexent model, widely used SDM.

However, there are few errors in citation and representation of results. In addition, there are redundancy in figures.

-    Errors in citation

The key reference (Li et al 2021) in material and methods is not found. Another reference (Liu et al 2020) is not also found. Please check carefully all citations and references.

-  Not coincidence between text and figure

In line 226, “13.0–23.45°C”. In Figure 5, however, minimum value is probably about 1-2℃. In line 364, suitable range is commented as “-2.01 to 24.92°C”. I do not understand this discordance.

In line 355-356, “the precipitation variation coefficient (bio15), were the main environmental factors affecting the distribution of M. destructor (Figure 3)”. In Figure 3, however, tmax 6 is more important than bio 15. Importance of tmax 6 should be discussed.

-        Redundancy in figures

Table 2 and Figure 9 is same. Hence, Figure 9 should represent percent of suitable area as text. This is the same in Table 3 and Figure 11.

In Figure 8, the sentence “The probability of the presence of the M. destructor is shown by the color scale in the area: red indicates a high-suitable area, yellow indicates a moderate-suitable area,

blue indicates a low-suitable area, and white represents an unsuitable area.” is not needed. The four

type areas are well explained within the figure. This is the same in Figure 10.

-        Other errors or suggestion Line 278: Figure b --à Figure 9b

Line 218-219: The sentence “which is an indicator that measures the variation in the total monthly precipitation throughout the year” should be deleted. Readers would misunderstood that this sentence represent 4 variables (tmax-11, tmin-2, prec-3, and bio-15).

In 45, “niches”. Niche is species-characteristic environmental conditions, and related with long-term evolution. Hence, distribution rather than niches might be proper.

Reviewer 2 Report

Prediction of the Current and Future Distributions of the Hessian Fly, Mayetiola destructor (Say), under Climatic Change in China

Comments:

This manuscript is well organized, well written, and addresses an important topic.  The methodology used is well established, as is the general interpretation of the results from this type of research.  I have only one general comment to offer.  I would suggest that the study does not offer a theoretical basis for management of the Hessian fly (lines 33-34; 109-110; 115-117), but, rather, an empirical basis.  (Also, the sentence on lines 116-117 is redundant with the previous sentence and could be deleted.)

A few minor editorial details caught my eye, but I hasten to add that I did not read the paper with detailed editing in mind.

Line 65: “zero” instead of “zeros”

Line 99: delete “widely”

Line 123: “destructor” instead of “destructo

Line 181: “al., 2021” instead of “al. 2021”

Line 232: delete “line”

Line 247: delete “in the area”

Lines 259-260: delete “in the area”

Line 278: “(Figure 9b)” instead of “(Figure b)”

Line 290: delete “in the area”

Line 293: “future climate scenarios” instead of “conditions”

Line 331: delete “in the area”

Line 359: “suggesting” instead of "proving”

Line 444: “study” instead of "experiment”

Lines 464-465: “scenarios” instead of "experiment”

Line 472: “are” instead of "is”
